# Current State of Pediatric Cardio-Oncology: A Review

**DOI:** 10.3390/children9020127

**Published:** 2022-01-19

**Authors:** Molly Brickler, Alexander Raskin, Thomas D. Ryan

**Affiliations:** 1Children’s Wisconsin, Milwaukee, WI 53226, USA; mbrickler@chw.org; 2Cincinnati Children’s Hospital Medical Center, Cincinnati, OH 45229, USA; thomas.ryan@cchmc.org

**Keywords:** cancer, cardio-oncology, cardiovascular, oncology, pediatric

## Abstract

The landscape of pediatric oncology has dramatically changed over the course of the past several decades with five-year survival rates surpassing 80%. Anthracycline therapy has been the cornerstone of many chemotherapy regimens for pediatric patients since its introduction in the 1960s, and recent improved survival has been in large part due to advancements in chemotherapy, refinement of supportive care treatments, and development of novel therapeutics such as small molecule inhibitors, chimeric antigen receptor T-cell therapy, and immune checkpoint inhibitors. Unfortunately, many cancer-targeted therapies can lead to acute and chronic cardiovascular pathologies. The range of cardiotoxicity can vary but includes symptomatic or asymptotic heart failure, arrhythmias, coronary artery disease, valvar disease, pericardial disease, hypertension, and peripheral vascular disease. There is lack of data guiding primary prevention and treatment strategies in the pediatric population, which leads to substantial practice variability. Several important future research directions have been identified, including as they relate to cardiac disease, prevention strategies, management of cardiovascular risk factors, risk prediction, early detection, and the role of genetic susceptibility in development of cardiotoxicity. Continued collaborative research will be key in advancing the field. The ideal model for pediatric cardio-oncology is a proactive partnership between pediatric cardiologists and oncologists in order to better understand, treat, and ideally prevent cardiac disease in pediatric oncology patients.

## 1. Introduction

The landscape of pediatric oncology has dramatically changed over the course of the past several decades, with five-year survival rates surpassing 80%. Improved survival has been in large part due to advancements in chemotherapy, refinement of supportive care treatments, and development of novel therapeutics, such as chimeric antigen receptor T-cell therapy (CAR-T) and immune checkpoint inhibitors (ICI) [1,2,3]. However, with improved survival rates, a five- to six-fold increase in cardiovascular disease risk has been observed, and cardiovascular disease is now the leading non-cancer cause of death. The range of cardiotoxicity can vary, but includes symptomatic or asymptotic heart failure, arrhythmias, coronary artery disease, valvar disease, pericardial disease, hypertension, and peripheral vascular disease [4,5,6]. Many patients will be asymptomatic for prolonged periods and may present for care at a late stage of disease if not appropriately screened early.

### 1.1. Mechanisms of Cardiac Toxicity

#### 1.1.1. Conventional Chemotherapy

Anthracycline therapy has been the cornerstone of many chemotherapy regimens for pediatric patients, since its introduction in the 1960s. Cardiotoxicity is the main dose limiting side effect that was reported a decade after its first use [7,8,9,10]. While the mechanisms of anthracycline cardiotoxicity are multifaceted, one key pathway is through interaction with topoisomerase 2β, which leads to nuclear DNA damage, mitochondrial dysfunction, and formation of reactive oxygen species [11,12,13,14]. Acute onset cardiotoxicity caused by an anthracycline is rare; it is defined as occurring within one week of administration of the anthracycline and is often reversible with discontinuation. Early onset chronic cardiotoxicity occurs within one year of administration, and late onset chronic cardiotoxicity presents greater than one year after administration of an anthracycline. For the chronic forms, disease is generally progressive. The current definition of high dose anthracycline exposure within the Children’s Oncology Group (COG) is a doxorubicin equivalent of 250 mg/m^2^. However, there are reports of children developing cardiovascular disease with doses as low at 60 mg/m^2^ [15,16]. Importantly, recent data have demonstrated that certain accepted dosing equivalencies for mediations like mitoxantrone may actually underestimate the cardiotoxic effect of such therapeutics on survivors of childhood cancer [17].

Non-anthracycline chemotherapy agents are not always thought of as cardiotoxic. However, there is a growing body of evidence demonstrating that alkylating agents (e.g., cyclophosphamide), microtubule inhibitors, proteasome inhibitors, platinum-based drugs, and antimetabolites contribute to cardiovascular disease, which can manifest as ventricular dysfunction, ischemia, venous thromboembolism, arrhythmia, and QT prolongation (Table 1) [18]. Therefore, all patients undergoing cancer therapy are at an increased risk of developing cardiotoxic side effects, regardless of the treatment modality utilized [19,20,21,22].

#### 1.1.2. Radiation

Radiation therapy is associated with cardiotoxicity through direct or indirect exposure of cardiovascular structures to the radiation field, dependent on the type and location of a cancer [23]. This is likely due to the initiation of an inflammatory cascade, generation of fibrosis, and development of endothelial dysfunction. Clinically, this may manifest as pericardial disease, coronary artery disease, calcification of the aortic root, conduction system abnormalities, valvar tissue injury (in severe cases leading to aortic and mitral valve stenosis), cerebrovascular disease, peripheral vascular disease, and heart failure [6,24,25]. While there is no known safe dose of radiation, high risk radiation has been accepted as >30 Gy of total exposure and >15 Gy for direct cardiac exposure [26]. The reduction in exposure to cardiac radiation from the 1970s to the 1990s has led to a significant decrease in heart failure and late-term coronary artery disease in adult survivors of pediatric cancers [27]. Alternatively, utilizing proton therapy may spare the heart from radiation exposure, which could in turn reduce the risk of cardiotoxicity [23]. There is literature suggesting that different areas and structures of the heart are able to withstand varying amounts of total doses of radiation before overall clinical change is seen. Novel research will be needed to advance this area, to further reduce cardiotoxity as well [25].

Intensity-modulated radiation therapy is a novel approach to directing radiation effect to desired fields (i.e., neoplasms), while sparing unaffected tissue. A recent prospective clinical trial determined that whole lung radiation, using intensity-modulated radiation therapy, with the goal of sparing the cardiac field, was feasible and offered similar cancer outcomes but lower doses of total Gy to the heart [28]. These data will be incorporated into the next generation of the Children’s Oncology Group Wilms Tumor Clinical Trials [29]. These studies will allow for the long-term follow-up of both oncologic and cardiac effects, to determine efficacy and safety.

#### 1.1.3. Chimeric Antigen Receptor T-Cell Therapy

The advent of CAR-T therapy has increased remission rates for refractory or relapsed acute lymphocytic leukemia. CAR-T therapy utilizes genetically engineered T-cells to target specific cancer antigens [30,31,32]. However, a major, and potentially fatal, complication of CAR-T therapy is cytokine release syndrome (CRS). This syndrome is defined by a triad of fevers, hypotension, and hypoxia with multi-organ involvement, driven by high levels of inflammatory cytokines (IL-6, TNF-alpha, IL-10, and IFN-Y). CRS can range in severity from mild to severe. This cascade of events can lead to cardiovascular dysfunction, including tachycardia, heart failure, and even death. The mechanism of action is unclear, but it is hypothesized that IL-6 plays a role similar to that in sepsis-related cardiomyopathy. This is confounded by patients having previously received cardiotoxic therapies such as anthracyclines and radiation prior to undergoing treatment with CAR-T therapy. However, the timing of a cardiac event following CAR-T is somewhat predictable, with most occurring just under a week following the infusion [33,34,35,36].

It is imperative that clinicians monitor for CRS and be conscientious that CRS may lead to serious and devastating cardiovascular injuries. Echocardiogram, ECG, and cardiac biomarkers should be obtained in the setting of progressive and severe CRS. Treatment with Tocilizumab, an anti-IL 6 receptor antagonist, may reverse CRS and prevent long-term cardiovascular complications [37]. Inotropic and vasoactive agents may be required to support patients with ventricular dysfunction and hypotension. Unfortunately, ventricular dysfunction can persist after CAR-T cell therapy [38,39].

#### 1.1.4. Immune Checkpoint Inhibitors

Therapy with ICI has changed the landscape of cancer treatment. In recent years, ICI have become more prevalent and at times are used as a front line therapy for a subset of pediatric cancers, which has helped to advance cure rates [40,41]. By definition, ICI are monoclonal antibodies that alter the patient’s immune response to cancer, leading to cell blockade or apoptosis. Cytotoxic T-lymphocyte associated antigen-4 (CTLA-4) and the programmed cell protein-1 (PD-1) pathways are the two most common targets utilized with ICI [42]. Both of these pathways are critical in T-cell regulation; thus, altering the pathway has autoimmune side effects that may affect every organ [43].

Cardiac related adverse effects of ICIs are becoming more frequent, and it is necessary to monitor patients closely throughout therapy. The most common cardiac related events are myocarditis, pericarditis, vasculitis, and arrhythmias. While myocarditis from ICI is a rare occurrence, it has a mortality rate of 40%. This high rate of mortality is thought to be a direct result of T-cell dysregulation targeting the heart, which can translate into clinical symptoms of arrhythmias, congestive heart failure, pneumonitis, or myositis. Vasculitis fatality rate is reported at 6% and can manifest with arthritis and rashes [44].

#### 1.1.5. Small Molecule Inhibitors

Targeted cancer therapies include tyrosine kinase inhibitors, vascular growth factor inhibitors, human epidermal growth factor-2 targeted therapies, and platelet-derived growth factor inhibitors. Currently, over 20 small molecule tyrosine kinase inhibitors, such as sorafenib and imatinib, are available for clinical use [45,46,47]. These drugs are well suited for cancer therapy, given their impact on cellular proliferation, differentiation, and survival, particularly in malignancies. Tyrosine kinase inhibitors inhibit cancer cell proliferation by competing through ATP binding sites, thereby reducing the tyrosine kinase phosphorylation leading to cell dysregulation [48,49,50,51]. Their benefits in pediatric cancers have been established [52,53,54]. Although these medications affect tyrosine kinase pathways in the myocardium, there are currently limited data on their role in cardiotoxicity. Kinase inhibitors have been associated with ventricular dysfunction, hypertension, pulmonary hypertension, and thromboembolism [55,56,57,58].

#### 1.1.6. Targeted Antibody Therapy

Targeted antibody therapy can be used to disrupt molecular pathways, similarly to those affected by small molecule kinase inhibitors [59,60,61]. Given the overlap in the mechanism of action, these monoclonal antibodies can produce similar cardiotoxic effects [62]. Trastuzumab has a primary role in the treatment of human epidermal growth factor 2 positive (HER2+) breast cancers. It is a murine monoclonal antibody that stops the proliferation of overexpression of HER2+ cells. It has been found to work in synergy with many traditional chemotherapy agents commonly used to treat breast cancer [63]. While it is currently the standard of care in this patient population, the most frequent adverse outcome with use of trastuzumab is cardiotoxicity [64]. In pediatric patients, antibody therapies have demonstrated effectiveness in various cancers [59,60,61,65]. These monoclonal antibodies can also cause cardiotoxicity, but clinical trials have shown that standard heart failure therapies offer protection [66].

### 1.2. Cardioprotection and Prevention

#### 1.2.1. Alternative Anthracycline Dosing Strategies and Derivatives

To mitigate anthracycline mediated cardiotoxicity, alternative dosing strategies have been utilized. In adult patients, increasing anthracycline infusion duration to longer than 6 hours may reduce the risk of subclinical cardiac injury, when compared to a shorter time of administration. Due to limited data in pediatric patients, these data cannot be extrapolated to this population. There was no difference in cardiotoxicity seen for those receiving single peak doses >60 mg/m^2^ of doxorubicin, when compared to <60 mg/m^2^ [67]. A liposomal formulation of doxorubicin was developed in order to allow clinicians to overcome the lifetime cumulative dose maximum. This formulation encapsulates the doxorubicin with a phospholipid bilayer of methoxypolyethylene glycol [68]. Liposomal doxorubicin allows a longer half-life, but with decreased cardiac side-effects, and is the only derivative definitively shown to decrease cardiotoxicity [69].

#### 1.2.2. Dexrazoxane

Dexrazoxane (Zinecard) is an EDTA derivative that acts as an iron chelator. It was first approved by the U.S. Food and Drug Administration in 1991 for prevention of cardiomyopathy associated with doxorubicin in breast cancer patients. In 2014, it was designated as an orphan drug for prevention of cardiomyopathy in pediatric and adolescent patients receiving anthracycline therapy. Dexrazoxane has been shown to be cardio-protective by multiple groups. Its primary mechanism of action is to prevent mitochondriopathy by chelating myocardial iron, preventing it from coupling with anthracyclines and reducing the formation of superoxide free radicals. Data support the cardioprotective effects of dexrazoxane, as manifested by improved troponins, natriuretic peptides, and function by echocardiography [70,71,72,73]. Despite the potential benefit, dexrazoxane has not routinely been utilized, due to concerns over its impact on anthracycline treatment effect and the risk of secondary malignancies. The risk of secondary malignancy or decreased efficacy of anthracyclines against the primary cancer are reasons dexrazoxane has not been widely incorporated into pediatric care. Some studies have shown dexrazoxane to be safe in these regards, [74,75,76,77], while others suggest a statistically borderline increase in risk [78]. COG now mandates the use of dexrazoxane in children who have a life time cumulative dose greater than >150 mg/m^2^ of anthracyclines or any dose of anthracyclines with concomitant radiation use [79].

#### 1.2.3. Exercise and Modifiable Risk Factors

Patients who have cancer and undergo cancer treatment are more likely to have modifiable cardiovascular risk factors, such as hypertension, diabetes, and obesity. Importantly, pre-existing cardiovascular risk factors are strong predictors for development of anthracycline- and radiation-related cardiotoxicity [80]. In addition, the incidence of medical frailty, as defined by five domains (walking limitations, low energy, exhaustion, low lean mass, and weakness) is significantly higher in survivors of pediatric cancer than in sibling controls [81]. Structured exercise demonstrates improvement in mortality, cancer progression, cancer recurrence, health-related quality of life, cardiovascular risk factors, and frailty in a dose-dependent manner [82,83,84,85,86,87]. Routine exercise in adults has been shown to improve cardiovascular function, immune function, body composition, chemotherapy completion rates, and reported markers of mental health. Several studies in adult survivors of pediatric cancers, as well as limited studies in pediatric patients, have likewise demonstrated decreases in cardiovascular-related and total mortality, often in a dose-dependent fashion [85,88,89]. As such, the American Cancer Society has established the ‘Moving Through Cancer’ initiative, with the mission ‘to ensure that that all individuals living with and beyond cancer are assessed, advised, referred to, and supported to engage in appropriate exercise and rehabilitation programming as the standard of care.’ [90].

Aerobic activity is generally considered safe for survivors of pediatric cancer and is advised as part of a ‘heart healthy lifestyle’. Traditionally, patients were advised to avoid isometric/weightlifting activities. However, recent guidelines from the National Comprehensive Cancer Network include recommendations regarding strength training activity for patients with normal ventricular function. There are exercise guidelines for cancer survivors published by the American College of Sports Medicine, although these are primarily adult focused [91]. COG only cautions against such activities for individuals with ventricular dysfunction. For patients who wish to participate in competitive sports, standard guidelines for athletic participation should be followed and ongoing monitoring by a cardiologist is recommended. In 2019 the American Heart Association released a Scientific Statement about cardio-oncology rehabilitation exercise (CORE) programs, including a safety checklist prior to engaging in CORE, components of CORE, and recommendations on how a patient should engage with various rehabilitation services [80]. Finally, clinicians should consider performing an assessment of physical activity when a patient is seen and provide an ‘exercise prescription’ that is safe and effective [92]. A systematic review demonstrated that adherence to such recommendations was improved by goal setting and instruction on how to perform the activities, and there were only a small number of adverse events [93]. Exercise interventions by telehealth have also shown good compliance and limited adverse events [94].

#### 1.2.4. Other Cardioprotective Strategies under Investigation

Remote ischemic conditioning using intermittent limb ischemia-reperfusion is a novel approach in the cancer community. Animal models demonstrated significantly reduced anthracycline cardiac toxicity with utilization of remote ischemic preconditioning [95,96,97]. Currently, clinical studies are ongoing to assess the feasibility and efficacy of remote ischemic conditioning in humans [98].

The COG ALTE1621 study is a multi-center, prospective, randomized, placebo-controlled trial intended to determine if low-dose carvedilol can prevent left ventricular remodeling and dysfunction in survivors of pediatric cancer. The goal enrollment is 250 individuals diagnosed at <21-years-old and treated with high-dose anthracyclines (>300 mg/m^2^), who will be followed for a period of 2 years. Participants will undergo scheduled assessments with echocardiographic and serum biomarkers [99].

## 2. Screening and Surveillance

Childhood cancer survivors are a unique group of patients, who require a collaborative approach to optimize their care. COG has published survivorship guidelines that provide broad health counseling for potential late side effects, including carotid artery disease and cardiac toxicity (cardiomyopathy, heart failure, and valve disease), with referral to Cardiology if concerns arise.

### 2.1. Risk Prediction

Data from the Childhood Cancer Survivor Study (CCSS) and other studies identified factors that increase the risk of developing cardiac toxicity including: younger patient age, African American race, female sex, total anthracycline dose, concomitant radiation exposure, underlying heart disease, pre-modern radiation protocols, and time since treatment (Table 2) [5,100].

According to the National Comprehensive Cancer Network (www.nccn.org accessed 10 October 2021), patients that have undergone cancer treatment should be considered American College of Cardiology/American Heart Association stage A heart failure (no structural abnormality, but at risk to develop heart failure) [101]. Based on both patient and treatment risk factors from the CCSS data (Table 2), an online risk calculator (https://ccss.stjude.org/tools-documents/calculators-other-tools/ccss-cardiovascular-risk-calculator.html accessed on 10 October 2021) was created to predict risk of heart failure, ischemic heart disease, and stroke by age 50 years in survivors of pediatric cancers [102,103,104]. No specific surveillance or treatment recommendations are made by this risk calculator.

### 2.2. Surveillance Guidelines

Most of the existing guidelines regarding monitoring for the development of cardiotoxicity are established for adult patients, with limited discussion of adult survivors of pediatric cancer [105,106,107,108]. Unfortunately, there are no standardized guidelines for pediatric patients during therapy, and there are variations between protocols [67]. Adult studies revealed that up to 10% of patients can develop subclinical and asymptomatic ventricular dysfunction during induction therapy with anthracycline administration. A delay in recognition and initiation of treatment of just 1–2 months may produce adverse long-term outcomes [109,110]. Recent data in pediatric patients undergoing treatment for acute myeloid leukemia showed that early cardiac toxicity was significantly associated with reduction in event-free survival and overall survival over a 5-year follow up [111].

COG has produced surveillance recommendations for patients that have completed cancer therapy (www.survivorshipguidelines.org accessed on 10 October 2021). These guidelines recommend an annual history and physical exam; lab work, including a lipid profile and glucose every 2 years; ECG during initial evaluation and then as necessary; and echocardiogram every 2–5 years based on risk factors. There are currently no recommendations for pediatric-specific imaging protocols or recommendations regarding the use of serum biomarkers, although efforts to develop such guidelines are underway. In addition to COG, other organizations have also created surveillance guidelines for childhood cancer (Table 3).

In summarizing and interpreting several separate recommendations, the International Late Effects of Childhood Cancer Guideline Harmonization Group recommends screening of left ventricular function with echocardiography as the preferred method, no later than 2 years after completion of anthracycline and/or radiation therapy. Repeat ECG and echocardiogram are recommended every 5 years thereafter, unless dictated otherwise by clinical status. More frequent and lifelong screening can be considered in high-risk survivors [113].

More recently developed imaging modalities such as 3-D echocardiography and myocardial strain assessment have been found to be more sensitive in identifying myocardial changes prior to changes of ejection fraction and shortening fraction [3,114,115,116]. Studies in adult cohorts have assessed the overall utility of myocardial strain in the patient with cancer, with ongoing debate as to the benefits of early detection of ventricular dysfunction weighed against the risk of modifying therapy for a change in strain when the ejection fraction remains normal [117,118]. Cardiac MRI or radionuclide angiography may be reasonable when echocardiography is not technically feasible or optimal. Cardiac MRI is used frequently in pediatric centers, while radionuclide studies are much less common than in adult patients. Cardiac biomarkers may be incorporated in conjunction with imaging but should not be used in isolation. Modalities such as stress echocardiography, exercise testing, and ambulatory rhythm monitoring are not included in the guidelines but are often considered and utilized based on clinical needs.

## 3. Therapeutic Approaches

### 3.1. Medical Heart Failure Therapy

Treatment of adult patients who develop heart failure should be directed by standard heart failure guidelines and supplemented by cardio-oncology-specific guidelines regarding changes in imaging, serum biomarkers, symptoms, and chemotherapy exposure risk stratification [119]. Standard medical management in adults includes use of ACE inhibitors, beta blockers, and statins [106,109,120,121]. Starting therapy in the first few months after the development of ventricular dysfunction can lead to improvements in systolic function in the vast majority of patients [109,110]. There are limited comparable data in the pediatric population. Guidelines for management of pediatric cardiomyopathy and heart failure exist, but they do not specifically discuss the cardio-oncology population [122]. ACE inhibitors can decrease left ventricular wall stress and improve subclinical markers of cardiac dysfunction in children. However, the long-term therapeutic effects are unclear [123,124]. Moreover, Lipshultz suggests that the long-term phenotype in survivors of pediatric cancer is that of ‘inadequate ventricular mass with chronic afterload excess associated with progressive contractile deficit and possibly reduced cardiac output and restrictive cardiomyopathy’, the so-called ‘Grinch syndrome’ in which treatment with an ACE inhibitor may be inappropriate [125]. A Cochrane database in 2016 showed no improvement in survival or development of heart failure in the limited number of studies that looked at various treatments, including one of enalapril with 135 survivors of pediatric cancers with asymptomatic LV dysfunction [126]. Sacubitril-valsartan (Entresto) has been studied and shown to have benefit in adult patients with cardiotoxicity, but it has not yet been studied in pediatric patients for this purpose [127].

Two recent surveys of practitioners who care for pediatric cardio-oncology patients found that the majority (>80%) use ACE inhibitors to treat ventricular dysfunction. Conversely, the addition of beta blockers varied between the two studies, with one survey reporting a 20% utilization rate and the other study up to 70%. Only one of the studies reported on the use of aldosterone antagonists, at approximately 50% [128,129].

Once therapies are started, it is unclear if and when they can be discontinued if function returns to normal. The TRED-HF study demonstrated higher rate of relapse of ventricular dysfunction after cessation of medical therapy when compared to patients maintained on medication [130]. Unfortunately, there are not stronger data to suggest a universal practice in this regard. Discontinuation of therapy should be made on an individual basis with the understanding that function may deteriorate.

### 3.2. Implantable Cardiac Defibrillators and Cardiac Resynchronization Therapy

Indications for an implantable cardiac defibrillator and cardiac resynchronization therapy are similar to other disease processes that cause heart failure and cardiomyopathy. Adult patients with cancer are less likely to receive an implantable cardiac defibrillator compared to other heart failure patients. There is a paucity of data on the efficacy of cardiac resynchronization therapy in cancer survivors, particularly involving pediatric patients [131].

### 3.3. Advanced Heart Failure Therapy

For some patients, standard oral therapies may become insufficient for the management of cancer treatment-mediated heart failure. In those cases, inotropic infusions, mechanical circulatory support, and even heart transplantation could be considered.

### 3.4. Heart Transplantation

The first reports of transplantation for anthracycline-induced heart failure in pediatric patients date back to the early 1990s [132,133]. Based on the International Society for Heart and Lung Transplantation guidelines, listing criteria for heart transplantation in patients with a cancer diagnosis should take into account a variety of factors, including type of neoplasm, response to therapies, risk of recurrence, and presence or absence of metastases. Active neoplasm and ongoing cancer treatment with chemotherapy/radiation are absolute contraindications to transplantation at most centers [134]. There is no defined time from the onset of remission to listing for heart transplantation.

For appropriately selected pediatric patients, there is no difference in long-term outcomes after transplant when compared to dilated cardiomyopathy [135,136]. In patients transplanted after a primary oncological diagnosis, there is concern for disease recurrence or increased risk of secondary cancers, related to the immunosuppression necessary to maintain a transplanted heart. However, data from the Pediatric Heart Transplant Society, representing 1985 transplants, demonstrated that all malignancies were due to post-transplant lymphoproliferative disorder, with no difference in malignancy rates in anthracycline-induced cardiomyopathy recipients [137,138,139].

### 3.5. Mechanical Circulatory Support

Patients that cannot wait for transplantation or are inappropriate for listing due to ongoing cancer therapy may be candidates for mechanical circulatory support. Short- or medium-term support strategies can be utilized as a bridge to recovery in the setting of temporary or reversible dysfunction [140]. Long-term support can be used as a bridge to transplantation or as destination therapy for those that are not transplant candidates. Several case reports and two small cohort studies have described the use of a left ventricular assist device as a bridge to recovery in adult patients with anthracycline-induced cardiomyopathy. Survival was similar to other causes of ventricular dysfunction. However, in one study, there was a higher need for subsequent right ventricular support in the anthracycline-induced cardiomyopathy group [141,142,143,144]. Data in pediatric patients are currently limited to a single case report [145].

## 4. Conclusions

Many cancer-targeted therapies can lead to acute and chronic cardiovascular pathologies. Unfortunately, there is lack of data guiding primary prevention and treatment strategies in the pediatric population, which leads to substantial practice variability. Several important future research directions have been identified, including those related to cardiac disease, prevention, management of risk factors, risk prediction, early detection, and the role of genetic susceptibility in development of cardiotoxicity. A combination of cohort studies and randomized controlled trials will be key in answering these important questions [3]. The future state within pediatric cardio-oncology should shift to a more proactive stance, to promote continued partnership between pediatric cardiologists and oncologists, in order to better understand, treat, and, ideally, prevent cardiac disease in pediatric oncology patients. Collaboration amongst specialties and across centers will provide critical data to further advance the rapidly growing field of cardio-oncology.

## Figures and Tables

**Table 1 children-09-00127-t001:** Cancer therapies associated with cardiovascular toxicity.

Treatment Agent	Potential Cardiovascular Toxicity
Anthracyclines	Ventricular dysfunction/heart failure
Radiation	Ventricular dysfunction/heart failure Valvular disease Pericardial disease Ischemic vascular disease/coronary artery disease Arrhythmias
Tyrosine kinase and Vascular endothelial growth factor inhibitors	Ventricular dysfunction/heart failure Hypertension Pulmonary hypertension Ischemic vascular disease/coronary artery disease Thromboembolism QT Prolongation
HER2-targeted agents	Ventricular dysfunction/heart failure
Immune checkpoint inhibitors	Myocarditis Arrhythmia
CAR-T cell therapy	Ventricular dysfunction Cytokine release syndrome-related hypotension
Alkylating agents	Ventricular dysfunction/heart failure Thromboembolism
Platinum-based agents	Ventricular dysfunction/heart failure Ischemic vascular disease/coronary artery disease Thromboembolism
Proteasome inhibitors	Ventricular dysfunction/heart failure
Antimetabolites	Ischemic vascular disease/coronary artery disease
Microtubule inhibitors	Arrhythmia Ischemic vascular disease/coronary artery disease
Other Thalidomide and analogs arsenic	Arrhythmia; Thromboembolism QT Prolongation

**Table 2 children-09-00127-t002:** Patient and treatment risk factors in the development of cancer treatment-related cardiotoxicity in patients treated for pediatric cancer.

Risk Factors	
Patient-Related	Treatment-Related
Younger age (especially <5 years of age)	Total cumulative anthracycline dose **
Female gender	Chest radiation ***
African American race	Time since treatment
Trisomy 21	Pre-modern radiation protocols (before 1975)
Cardiovascular risk factors (hypertension, hyperlipidemia, diabetes, obesity)	Concomitant therapy with cyclophosphamide, bleomycin, vincristine, amsacrine, mitoxantrone, immunotherapy
Underlying heart disease (congenital heart disease, cardiomyopathy)	
Genetic factors *	

* Multiple genotypes identified as risk factors. ** Dose cut-off frequently cited as >250 mg/m^2^ doxorubicin equivalent. *** Dose cut-off frequently cited as >15–30 Gy chest radiation.

**Table 3 children-09-00127-t003:** Resources providing information and/or guidance for cardiovascular care of survivors of pediatric cancers.

Resource
American Heart Association Scientific Statement on Pediatric, Adolescent, and Young Adult Long-Term Survivors [6]
Children’s Oncology Group (www.childrensoncologygroup.org)
National Comprehensive Cancer Network (nccn.org)
Dutch Childhood Oncology Group [112]
Scottish Intercollegiate Guidelines Network (www.sign.ac.uk)
UK Children’s Cancer and Leukaemia Group (www.cclg.org.uk)
International Late Effects of Childhood Cancer Guideline Harmonization Group [113]

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
