# Peer review of "Current State of Pediatric Cardio-Oncology: A Review"

_children, 2022, doi:10.3390/children9020127_

Round 1

Reviewer 1 Report

Thanks to the authors for this comprehensive reivew of the pediatric cardio-oncology. The initial sections on the different treatment modalities could be further improved with some more details on how each may lead to cardiac dysfunction. Specifically, in the radiation section it would be helpful to distinguish between prescribed dose and the dose received by the heart, particulary cardiac substructures and how that can impact future toxicity. Further discussion around the use of smaller radiotherpay fields and IMRT to avoid cardiac toxicity would also be helpful. More details on how TKIs, trastuzumab and immunotherapy cause cardiac toxicity would also be helpful. 

A great addition to the article would be to give an idea of how many children are treated with the different therapies, what proportion then go on to develop cardiac problems and what the risk factors are with each therapy rather than just stating the overall risk factors for cardiac toxicity in general.

Author Response

Thank you for the feedback. We have incorporated more information regarding IMRT, radiation therapy, and mechanisms of novel cancer treatments. While we would love to include detailed numbers of patients treated with different therapies unfortunately data tracking for such a purpose is lacking. Hopefully, with time and greater organization between treatment centers that data will be readily available. 

Reviewer 2 Report

Review of pediatric cardio-oncology, comprehensive, well-founded, with adequate references.

On page 6 correct the repetition of words:the “Moving Through Cancer” initiative, with the mission “to ensure that ensure that the “Moving Through Cancer” initiative, with the mission “to ensure that ensure that 

Author Response

Thank you for this correction. The appropriate revision has been made.